# Exploring Response to Immunotherapy in Non-Small Cell Lung Cancer Using Delta-Radiomics

**DOI:** 10.3390/cancers14020350

**Published:** 2022-01-11

**Authors:** Emanuele Barabino, Giovanni Rossi, Silvia Pamparino, Martina Fiannacca, Simone Caprioli, Alessandro Fedeli, Lodovica Zullo, Stefano Vagge, Giuseppe Cittadini, Carlo Genova

**Affiliations:** 1Interventional Angiography, Ospedale Santa Corona-ASL 2 Savonese, 17027 Pietra Ligure, Italy; emanuele.barabino@gmail.com; 2UOC Oncologia Medica 2, IRCCS Ospedale Policlinico San Martino, 16132 Genova, Italy; lodozullo@gmail.com; 3Department of Medical, Surgical and Experimental Sciences, University of Sassari, 07100 Sassari, Italy; 4Department of Health Sciences (DISSAL), Ospedale Policlinico San Martino, University of Genova, 16128 Genova, Italy; pamparinosilvia@gmail.com (S.P.); fiannacca.martina@tiscali.it (M.F.); simone.caprioli11@gmail.com (S.C.); 5Department of Electrical, Electronic, Telecommunications Engineering, and Naval Architecture, University of Genova, 16145 Genova, Italy; alessandro.fedeli@unige.it; 6Department of Radiation Oncology, IRCCS Ospedale Policlinico San Martino, 16132 Genova, Italy; stefano.vagge@hsanmartino.it; 7UO Radiologia Generale, IRCCS Ospedale Policlinico San Martino, 16132 Genova, Italy; giuseppe.cittadini@hsanmartino.it; 8UOC Clinica di Oncologia Medica, IRCCS Ospedale Policlinico San Martino, 16132 Genova, Italy; carlo.genova@hsanmartino.it; 9Dipartimento di Medicina Interna e Specialità Mediche (DiMI), Facoltà di Medicina e Chirurgia, Università degli Studi di Genova, 16132 Genova, Italy

**Keywords:** radiomics, NSCLC, delta-radiomics, immunotherapy, antiPD1, immune checkpoint inhibitor, predictive value

## Abstract

**Simple Summary:**

The aim of the study is to identify radiomic features capable of predicting the response to immunotherapy. Delta-radiomics can foresees the comparison between subsequent CT scans and therefore allows to predict the changes that occurred during the treatment. In this study, the individual lesions of patients with advanced non-small cell lung cancer treated with immunotherapy were analyzed. The study aims to discover the features that predict the response to immune checkpoint inhibitors.

**Abstract:**

Delta-radiomics is a branch of radiomics in which features are confronted after time or after introducing an external factor (such as treatment with chemotherapy or radiotherapy) to extrapolate prognostic data or to monitor a certain condition. Immune checkpoint inhibitors (ICIs) are currently revolutionizing the treatment of non-small cell lung cancer (NSCLC); however, there are still many issues in defining the response to therapy. Contrast-enhanced CT scans of 33 NSCLC patients treated with ICIs were analyzed; altogether, 43 lung lesions were considered. The radiomic features of the lung lesions were extracted from CT scans at baseline and at first reassessment, and their variation (delta, Δ) was calculated by means of the absolute difference and relative reduction. This variation was related to the final response of each lesion to evaluate the predictive ability of the variation itself. Twenty-seven delta features have been identified that are able to discriminate radiologic response to ICIs with statistically significant accuracy. Furthermore, the variation of nine features significantly correlates with pseudo-progression.

## 1. Introduction

In radiomics, digital images are considered as minable data from which high-throughput features could be extracted [1]. Radiomics can describe radiological images beyond the perception of the human eye, and it has been hypothesized that this approach might individuate and quantify macroscopic radiological phenotypes that are the result of certain cellular or tissue characteristics [2]. These data are exceedingly difficult to handle, and they are usually processed with very complex mathematical methods, such as Machine Learning (ML), to create predictive models able to answer specific questions. However, an effective correlation of the radiomic features with their biological counterparts has not been proven [2]: hence, extrapolations on feature meanings remain speculative, which means we are creating elaborate predictive models using data that are not completely understood. As many attempts have been made to select the most reliable and less redundant features using methods such as LASSO, PCA [3], or test-retest [4], the effort to comprehend radiomic features remains an unaddressed challenge.

Delta-radiomics is a subspecialty of radiomics that studies the variations of the radiomic features in time [5] or after the introduction of an external agent either pharmacologic (chemotherapy, immunotherapy) or physical (radiotherapy) to produce biomarkers able to describe or predict a specific clinical situation (i.e., response, early recurrence). Delta-radiomics does not depict a static situation, but its evolution; confronting delta-radiomic features means comparing different starting situations that present similar changes. To date, radiomics has been used to predict definite biological (i.e., mutational status) or clinical (i.e., response to therapy, overall survival) conditions but has been rarely employed to perform a longitudinal assessment of disease evolution [6,7,8].

Since their introduction in research and clinical practice, immune checkpoint inhibitors (ICIs) have revolutionized the treatment of non-small cell lung cancer (NSCLC), by increasing patients’ survival both locally advanced and in the metastatic stage. However, while during chemotherapy, response criteria are purely dimensional (increase/decrease/stability) and the correlation between response and dimensions is linear, this relationship is not always as straightforward when ICIs are employed. Indeed, during treatment with immunotherapy, malignant lesions can show a large spectrum of heterogenous responses that vary in terms of form, timing, and duration; this heterogeneity in response forced the medical community to elaborate new and most suitable criteria to depict the changes determined by the immune response unleashed during immunotherapy such as iRECIST and irRECIST [9,10]. In some cases, the tumor significantly enlarges before shrinking, a response known as pseudo-progression (pPD); this occurrence is typically associated with the enrollment and activation of tumor-infiltrating lymphocytes. The unmet clinical need to describe the tumor response has even pushed research to perfect new radioactive biomarkers [11]. However, in Computed Tomography (CT), even the newest response criteria do not describe the interactions between tumor and immune system and are still dependent on linear dimensions. Radiomic features are derived from complex mathematical equations and, in some cases, are designed to be sensible to definite characteristics of an image; to date, few studies tried to reverse-engineering radiomic features to understand the characteristics of radiological images and biological processes. The aims of our study are: (a) to individuate radiomic features that might be used as surrogate biomarkers of response to immunotherapy in NSCLC (b) to employ delta-feature trends to describe the response to ICIs in NSCLC.

## 2. Materials and Methods

### 2.1. Eligible Patients and Radiologic Lesions

In this study, we included consecutive patients with cyto-histological diagnosis of advanced NSCLC, who were treated in first or second line with an anti PD-1 ICI (nivolumab or pembrolizumab). We included only patients for whom a baseline CT scan and the first response assessment were available in our archive. Both CT scans had to be performed with contrast medium. In this study, we considered only lung lesions, and the same lesion had to be present both at baseline and first re-evaluation (complete responses were excluded). The Consort diagram summarizes how we selected the radiological exams. Sixty-two patients were enrolled in the study; from these, 24 patients were excluded due to lack of reevaluation CT (early death, complete response of target lesion, absence of contrast medium in reevaluation CT). In addition, for five patients, there were problems during the process of reformatting of the slices. Therefore only 33 patients were considered, for a total of 43 target lesions (Figure 1). For the evaluation of the response, each lesion was considered separately. The response was assessed using the RECIST 1.1 cut-off criteria: greater than 30% decrease for partial response (PR), greater than 20% increase for progressive disease (PD), while stable disease (SD) was defined when previous conditions were not met. The complete response was not taken into consideration given the absence of the reassessment of the lesion.

Lesions that met the criteria for pPD were re-checked [12]. Although the planned radiomic analysis included the exclusive assessment of the baseline and the first re-evaluation CT scans, in order to generate a predictive model of response, we evaluated the response in the subsequent CT scans performed throughout the patient’s therapy. Finally, we assigned the best response available in subsequent patient evaluation (an SD at the first re-evaluation that became a PR after other re-evaluations were considered a PR). For the same reason, in case of disease progression, a SD at the first re-evaluation was considered a PD (Figure 2).

### 2.2. CT Protocol and Images Reformatting

All the patients underwent contrast-enhanced CT scans using the standardized protocol of our institution. CT acquisition was performed on two 16-rows and a 64-rows scanner (LightSpeed and Optima, Ge Healthcare, Chicago, IL, USA) after contrast media injection (80–120 mL, Iopamiro 370, Bracco) during the venous phase. The acquisition parameters were the following: tube voltage 120 kV, slice thickness 1.25 mm, FOV 35–50 cm, matrix 512 × 512. Tube current automatic modulation was employed. DICOM images were transferred on a dedicated workstation (AW Server 4.6, GE Healthcare, Chicago, IL, USA) and reformatted on the axial plane at 3.0 mm of slice thickness.

### 2.3. Segmentation and Radiomic Analysis

Reformatted images were transferred to an offline workstation. Segmentation was performed using 3D Slicer 4.10.1 [13]. A radiologist with six years of experience in radiomics, blinded to clinical and chronological data, delineated a Volume-of-Interest (VOI) around each lesion using a semi-automated method and a slice-per-slice approach. Hyperdense structures, such as bone or metal objects, were excluded from the segmentation; if present, the air was included within the VOI, even in case of cavitation at the consequent CT, but major lobar branches were excluded. Radiomic features were calculated with the Pyradiomics extension [14]. Ninety-three features were calculated for each lesion before (baseline) and after immunotherapy (first assessment). Radiomic features were extracted using the following methods: first-order statistics (*n* = 18), Gray-Level Co-Occurrence Matrix (*n* = 24), Gray-Level Run Length Matrix (*n* = 16), Gray-Level Size Zone Matrix (*n* = 16), Neighbouring Gray Tone Difference Matrix (*n* = 5) and Gray-Level Dependence Matrix (*n* = 14). The mathematical definitions of the adopted features are available for consultation on Pyradiomics’ website (https://pyradiomics.readthedocs.io/en/latest/features.html accessed on 18 November 2022.).

### 2.4. Delta-Radiomic Features Computation

Radiomic features were extracted from the baseline CT scan and from the CT scan of the first re-evaluation. Delta-radiomic features were calculated employing two different approaches: Absolute Difference (AD) and Relative Variation (RV).

In the AD case, the variation of each radiomic feature V (i.e., the delta-radiomic feature $\Delta V_{AD}$) was calculated as
(1)$$\Delta V_{AD}=V_{T0}-V_{T1},$$
where $V_{T0}$ represents the value of the feature at baseline CT scan, and $V_{T1}$ is the value of the same feature at the first assessment CT (after immunotherapy).

In the RV method, the variation was also normalized with respect to $V_{T1}$, i.e., computing the delta feature $\Delta V_{RV}$ as
(2)$$\Delta V_{RV}=\frac{V_{T0}-V_{T1}}{V_{T1}}.$$

### 2.5. Statistical Analysis

Delta features of the three classes of response (PD, PR and SD) were compared using ANOVA or Kruskal–Wallis tests [15,16]. Median and standard deviation of each delta feature were calculated. The statistical significance level was set at *p* = 0.05. Statistical analysis was performed using GraphPad Prism version 7.00 for Windows (GraphPad Software, La Jolla, CA, USA, www.graphpad.com accessed on 18 November 2021.) Due to the limited dataset, no machine learning or other type of artificial intelligence approach was adopted. Furthermore, the most important prognostic variables for NSCLC were evaluated by univariate analysis. Statistical significance between survival curves (Kaplan–Meier) was assessed using the Log-Rank Test and the Cox model. The variables considered for these analyses were histology (adenocarcinoma vs. squamous), Performance status according to Eastern Cooperative Oncology Group (ECOG < 2 or ECOG ≥ 2), and age (<65 years and >65 years). The PD-L1 value was not considered because all the patients receiving ICI in the first line had a tumor with PD-L1 expression ≥50%; conversely, PD-L1 expression value was not consistently available for the patients treated in the second line, apart from the two patients for whom it was available and higher than 50%. Therefore, this analysis was impossible to perform.

### 2.6. Radiomic–Radiological Correlation

At the end of data analysis, radiologist reviewed all the cases included in the study and unblinded data in order to establish a radiomic-radiological correlation between the features that are different in the three response classes (PD, PR and SD) and that can distinguish PD and pPD. In particular, the most extreme variations were taken into consideration in order to have the greatest probability of recognizing a variation visible to the human eye. Starting from these examples, it was possible to trace a certain variation to a known descriptive radiological pattern and from there, hypotheses were formulated about the meaning of these variations even when these are not visible to the human eye, but only traceable as a variation of the features.

## 3. Results

### 3.1. Patients

Globally, 33 patients were included in our study. Among these patients, 28 were male; median age was 67 years (range: 43–88). Seventeen patients received Nivolumab, while 16 received Pembrolizumab; 16 patients were evaluated during their first-line treatment, while 18 patients were treated as second-line. Globally, 10/33 patients were considered to have an ECOG performance status (PS) of 0, while 21 had ECOG PS 1, and only two patients had a ECOG PS of 2. The median Progression-Free Survival (PFS) was 6.5 months (3.8 for nivolumab group and 7.1 for pembrolizumab group), while the median Overall Survival (OS) was 13.7 months (9.3 for nivolumab group and 15 months for pembrolizumab group).

Forty-three pulmonary lesions were included in the analysis. The average time between the CT exams was 70.26 days, while the mean follow-up time was 11.5 months.

### 3.2. Features That Distinguish Response Classes

Twenty-seven delta features resulted in significantly different results in the three classes calculated by RV methods (Table 1). Considering the absolute difference (AD), 10 features were different in the three classes (Appendix A); furthermore, these 10 features were present among the 27 features identified with RV. Therefore, RV seems to offer the best descriptive and intuitive method to calculate delta features. Most of the features individuated with the RV method measure an increase/decrease in VOI heterogeneity. On the other hand, some features (GLRLM and GLSZM) are more composite and designed to be more sensitive to specific aspects of the VOI. The results for each class are presented in the following paragraphs.

#### 3.2.1. First-Order Features and Heterogeneity-Based Features

Most first-order, GLDM, GLCM and NGTDM features (“Energy”, “Total Energy”, “Contrast”, “GrayLevelNonUniformity”, “Coarseness”, “Strength” and “Busyness”) describe, in different ways, an increase or decrease of heterogeneity within the VOIs. All these features are concordant in describing an increasing homogeneity in PR and an increasing heterogeneity in PD of the voxels included in the VOIs. In PD, the feature Coarseness (*p* = 0.0062) and Busyness (*p* = 0.0205) describe a situation in which the VOI becomes more heterogenous and presents a higher spatial rate of change, meaning that variations are more abrupted.

The feature “Median” well discriminates PD from SD and presents negative values in PD patients and mildly positive values in SD patients, representing respectively an increasing and a slight decrease in the median attenuation value.

#### 3.2.2. Gray-Level Run Length Matrix Features

Gray-Level Run Length Matrix (GLRLM) features are designed to individuate and quantify consecutive pixels that have the same gray-level value within the VOI (Run-length or RL). “Run Length Non-Uniformity” (RLNU) is a feature created to quantify the homogeneity of RL in an image; an image with identical RL presents a very small value of RLNU. RLNU was significantly different in the three cohorts (*p* = 0.0036) but, after review, we understand that this difference is more evident in tumors near the pleural surface and that our observation could be determined by the involvement of pleurae and scissures (that, acting as a temporary barrier to tumoral invasion, create a prominent interface) or the presence of very prominent spicules; therefore, RLNU increasing/decreasing is determined by an augmented/reduced involvement of pleural surfaces or an enlargement of the spicules. The fact that GLRLM features are influenced by anatomical landmarks such as pleural surfaces highlights the possibility that, particularly in the lung, where tumors and lung parenchyma present a very different attenuation and create pronounced interfaces, these features are more sensitive to high-contrast areas, where the difference in attenuation creates homogenous, low density, linear interfaces. RLNU is significantly different in PD and PR patients: in PD the value is negative, meaning that the level of non-uniformity in the run-lengths increases, a phenomenon that could be explained by the increase, for example, of the spicules around the boundaries of the tumor that create an interface composed by multiple heterogeneous parts; conversely, in tumors that manifested PR the value of RLNU is reduced, which means that the boundaries of the tumor might have become less marked, more rounded, and linear, indicating a less infiltrative pattern.

#### 3.2.3. Gray-Level Size Zone Matrix

If we consider the features “Large Area High Gray-Level Emphasis” (LAHGLE Figure 3 and Figure 4) (which is designed to describe the preponderance within the VOI of coalesced regions with high-intensity pixel), we can appreciate how this feature can distinguish PR, SD and PD; in PR the delta-values of LAHGLE are positive, meaning that the feature that describes high-intensity regions have shrunk while in PD and SD such delta-values are negative, representing an increasing number and dimension of high-intensity regions. Assuming that in a lung lesions dataset, the densest pixels are those with contrast media, we can associate high-density areas with regions that present vivid uptake. By contrast, the feature “Large Area Low Gray-Level Emphasis” (LALGLE) describes the preponderance of large areas with low-density pixels; in PD and SD, LALGLE Δ decreases, which means that these are enlarged, and shrinks while in PR (Figure 3, Figure 4 and Figure 5).

### 3.3. Features That Distinguish PD and Pseudo-Progression

RV identified nine features that can distinguish pseudo-progression from progressive disease (Table 2). In particular, it is important to note that the features able to distinguish the progressive disease from pseudoprogression are different from those able to distinguish the responder from a lesion that will progress. Only two features have a statistically different delta both between PD and PR and between PD and pPD; these features are the “Run LengthNon Uniformity” and the “Gray-Level Non Uniformity”. This observation defines pPD as a completely different phenotype compared to actual PD. The values of Short Run Low Gray-Level Emphasis (SRLGLE) presented very different values in PD and pPD (*p* = 0.0007); SRLGLE is a feature that measures the joint distribution of shorter run lengths with lower gray-level values, with a higher value indicating a greater concentration of low gray-level values in the VOI. In PD, the value of SRLGLE increases, indicating the emergence of the small low-attenuated interface along the boundaries of the tumor such as spliculae; in pPD the value of SRLGLE decreases, meaning that the small interface reduced and globally, the margins of the tumor are less irregular and more rounded (Figure 6). Finally, in pPD the value of “Low Gray-Level Zone Emphasis” (LGLZE), a feature that quantifies the presence of lower gray-level values and size zones on a global level in the tumor, reduces significantly (Figure 5).

### 3.4. Univariate Analysis of Prognostic Clinical Factors

The three aforementioned prognostic factors (histology, ECOG PS, and age) were evaluated for all patients enrolled in the study. Regarding the histology, The HR in favor of the adenocarcinoma histology was 0.49 (CI95% 0.19–1.25, *p* = 0.06), not statistically significant. Similarly, the ECOG PS, the trend was in favor of patients with better ECOG PS with a HR of 0.46 (CI95% 0.22–0.97, *p* = 0.07), albeit statistical significance was not reached. Finally, considering younger patients versus older patients, the trend was in favor of younger patients with a non-significant HR of 0.86 (CI95% 0.42–1.77 *p* = 0.69) (Appendix A).

## 4. Discussion

The therapeutic revolution of immunotherapy in the treatment of NSCLC is widely acknowledged [17]. The search for predictive biomarkers of response has gathered the efforts of many researchers, without however leading to real advances in clinical practice so far. Tissue assessment of PD-L1 remains the best response biomarker available to date, despite its known limitations [18]. It has become increasingly clear that each tumor is not homogeneous; indeed, tumor heterogeneity is spatial (intra-lesion and inter-lesion) and temporal (as it can change over time) [19]. The tissue biopsy, on which we evaluate the expression of a biomarker, is only a part of the whole and returns an immutable result over time, which, therefore, does not respond to reality as well as an invasive diagnostic test for the patient. Research of circulating biomarkers and radiomics are the most promising ways to overcome these limitations. This work aimed to evaluate how the variation of the features can predict the future behavior of each lesion of a patient. Despite the sample size limits, that prevented the use of artificial intelligence algorithms, we were able to identify features that can predict the response of a lesion during immunotherapy treatment. This could be a successful approach, considering the consecutive and longitudinal evaluations of a patient allowing to discover in advance when a lesion will stop responding or, conversely, when it will become susceptible to response at any time during treatment. Radiomics is one of the main topics of research in the radiological field but, still, reproducibility represents the major issue that limits its translation in clinical practice. Compared to the number of papers on radiomics published in the last decade, delta-radiomics’ possibilities had been scarcely evaluated. Nonetheless, a study by Plautz et al., demonstrated that delta features are stable in phantoms [20] and a recent study by Nardone et al. [21] proposed that delta-radiomics can be used to improve multi-scanner and multi-center reproducibility.

Radiomic features calculated from the same tumor at different timepoints derive from an object acquired in similar conditions (i.e., position within the body, proximity to hyperdense structures and timing of arrival of the contrast media within the lesion); this process might eliminate much of the variability of radiomics. We can reasonably assume that in delta-radiomics, each patient is his/her own control throughout time, and changes are therapy-induced or determined by disease progression. Furthermore, delta-radiomics might open the path to deep insights within the response to immunotherapy. The analysis of radiomic features one by one demonstrated that something changes within tumors after the introduction of immunotherapy, and these observations are confirmed by the trend of significant delta features that is mostly different in amplitude and direction in the three classes. To our knowledge, only three studies established a relationship between delta features and response to immunotherapy [7,22,23] and none tried to interpret radiomic features and their variations. Our study demonstrated that 27 delta features presented a similar trend and might be associated with a definite response to immunotherapy. Moreover, we found nine features that might potentially distinguish pPD from actual PD.

Based on our findings, the features LAHGLE and LALGLE might describe disease progression, in which large, poorly perfused areas and variable contrast-enhanced regions enlarged at the same time. On the contrary, in responding lesions these areas, particularly enhanced ones, shrink and are reduced or not appreciable at the reassessment CT scan. A possible explanation of this fact is that progressing tumors could include larger, poorly perfused areas of necrosis. We know how necrosis, and therefore a hypovascularized microenvironment, constitutes by itself a tumor escape mechanism, with an increased production of HIF1 [24,25], which directs the tumor microenvironment towards an immunosuppressive phenotype. This condition might be described by the observed variations of LAHGLE and LALGLE (that could correspond to an enlargement of focal necrosis areas), which suggest how immunotherapy alone, in such cases, is not able to overcome this specific tumor escape. Notably, pPD presents values of LAHGLE similar to PD, while the value LGLZE reduces, indicating that, in spite of the persistence of low-density areas, the global attenuation of the tumor increases. Figure 5 is a graphical representation of these phenomena. This aspect, in accordance with the above, makes us hypothesize that increased capillary permeability to the contrast medium, which tends to increase the pixel density, is a positive predictive factor of response; on the other hand, the increased capillary permeability is known to be one of the first steps in the inflammatory response.

From a radiomic point of view, the most elusive class is SD; indeed, this class is very heterogenous and most of its values are halfway between the other classes. Nonetheless, the median value can be decreased in two ways: by eliminating high-attenuated pixels or by recruiting more hypodense pixels. The radiomic–radiologic review demonstrated that in most cases, this phenomenon regards both processes, as stable lesions become more homogenous, less enhanced and less attenuated compared to progressive ones.

Considering the methods of calculating the radiomic delta, another interesting aspect is that, between the two employed methods (AD, RV), the one that performed better is RV. This occurrence is not surprising; indeed, being a ratio, RV eliminates the variability of the features among the different lesions.

Radiomics presents undeniable reproducibility issues that are determined mostly by different acquisition parameters but also on the characteristics of the patient such as gender, weight and, more specifically, by the location of the tumor (i.e., tumors near bone, metal or air are more prone to beam hardening artifacts (Figure 7)). However, if we consider only the relative variations, most of these variables are eliminated, providing reliable delta-radiomic features. Moreover, RV values have a negative or positive value that indicates the variation’s trend, providing an intuitive and easily understandable biomarker.

We also evaluated the weight of the most important prognostic and predictive variables in NSCLC: age, histology, ECOG PS, and PD-L1 expression. For the last variable, it was not possible to perform a univariate analysis and this certainly falls within the limits of this work. Indeed, all the patients for whom the value was available had PD-L1 expression ≥50%, while the value was not available for most patients treated in the second line. For the other three variables, the limited sample size meant that none of them demonstrated statistical significance; this observation suggests that radiomic features might be better able to discriminate responding patients from non-responding patients compared to clinical features. Furthermore, the clinical characteristics are patient-dependent and not lesion-dependent; therefore, they are less informative if we seek the prediction of response lesion by the lesion.

Apart from the aforementioned lack of PD-L1 expression data, our study presents other limitations. In the first place, the cohort of patients was small due to the retrospective nature of the data and the need to comply with very strict inclusion criteria that would allow the radiomic analysis. This fact limits at the same time the strength of our results and the possibility to apply machine learning, AI-based, or other complex classification techniques. Moreover, the evolution of NSCLC treatment has witnessed the arrival of chemo-immunotherapy combinations, which are currently the best treatment available for a large proportion of patients; in our study, this combination was not considered for the sole purpose of having a simpler cohort to analyze. Another aspect that should not be underestimated is that the cohort includes both patients treated in the first line or in the second line. Finally, the effectiveness of our results should be tested on independent test sets to proper evaluate the reproducibility and generalizability of our findings.

## 5. Conclusions

In conclusion, delta-radiomic features might potentially provide reliable biomarkers of response in NSCLCs treated with immunotherapy. One of the most intriguing aspects is that, conversely to response criteria such as RECIST, delta-radiomics provides complex quantitative parameters that could better stratify patients and provide new information about intra-tumoral changes. Radiomic analysis requires no adjunctive exams, can operate using data from standard-of-care CT images and, therefore, is a cost-effective technique. Delta-radiomic features have the potential to overcome the limitations of iRECIST in immunotherapy and, possibly, quantify the response to treatment and eventually predict the course of therapy. The information provided by delta-radiomics might be employed to identify lesions that are not responding or that will no longer respond to immunotherapy in the short term, and that might be managed with a different approach (i.e., radiotherapy, local ablative therapy), leading to tailored treatments and actual precision medicine.

## Figures and Tables

**Figure 1 cancers-14-00350-f001:**
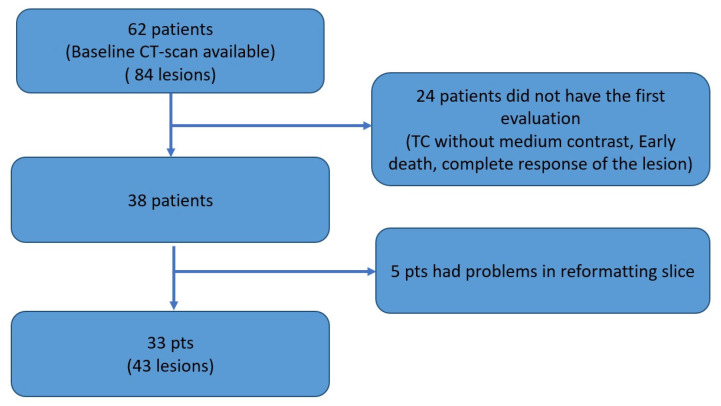
Consort diagram for selection of patients and lesions.

**Figure 2 cancers-14-00350-f002:**
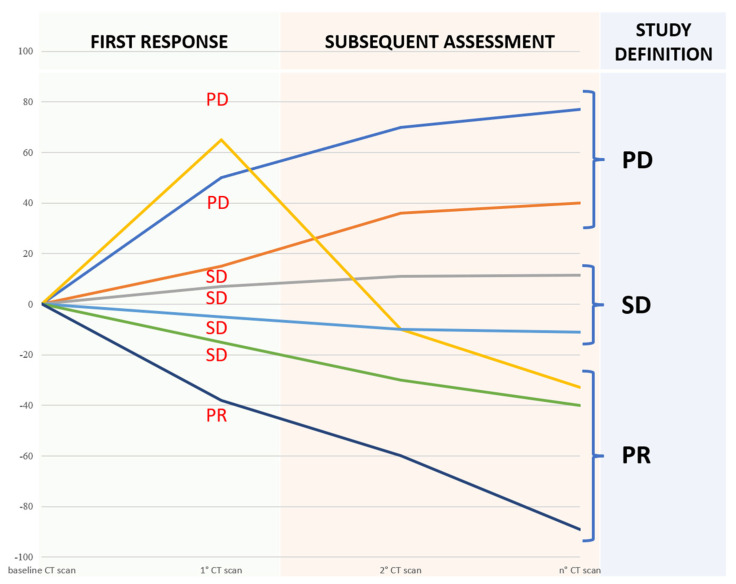
The best response known during subsequent assessment was adopted to define the actual response and understand the potential predictive role of delta-radiomics.

**Figure 3 cancers-14-00350-f003:**
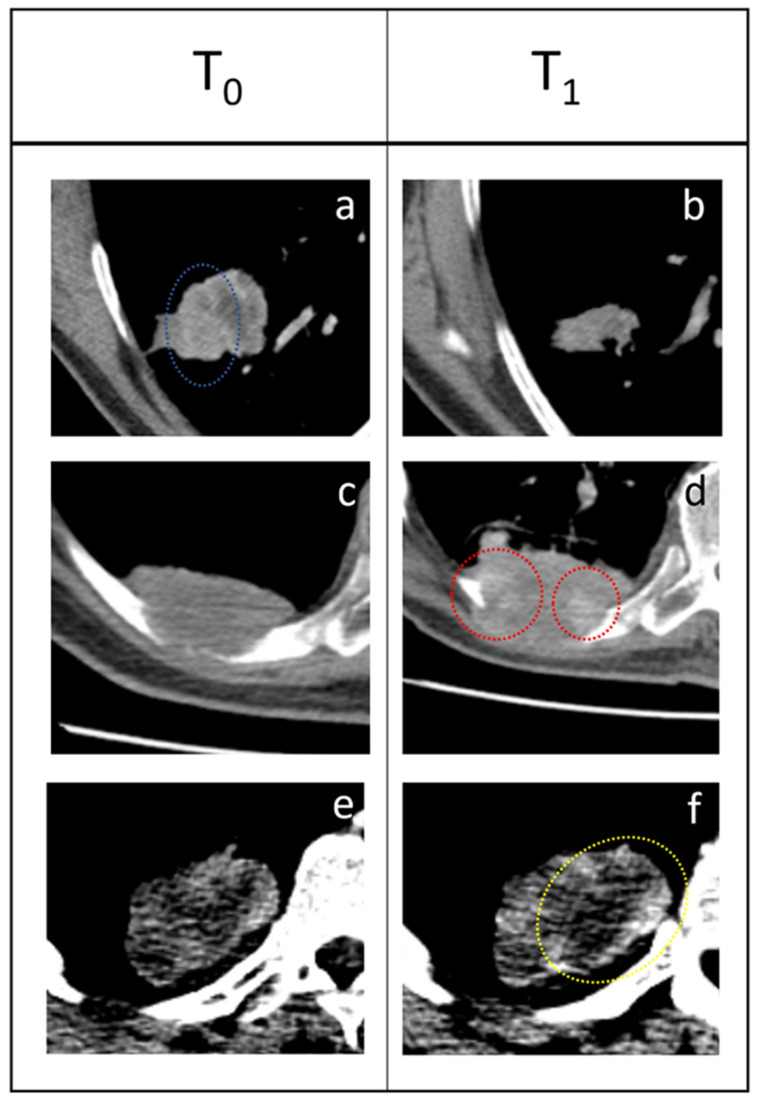
Three examples of response to immunotherapy. (**a**,**b**) PR: the tumor shrunk from T0 to T1 and areas that presented high enhancement (blue dotted circle) disappeared. (**c**,**d**) SD: 63-year-old male patient presenting a pulmonary tumor that penetrates the thoracic wall and erodes a rib with extensive bone reabsorption. The first re-evaluation showed an increased mass (53 × 35 mm vs. 50 × 25 mm) consistent with PD. However, during therapy, the lesion gradually reduced and became stable. Delta-radiomics individuated an extreme negative variation of LAHGLE (−0.999770105) that could be represented by the emergence of areas with vivid contrast media uptake (red dotted circles) within the lesion. (**e**,**f**) PD: this pulmonary lesion was slightly enlarged at the first follow-up but radiomic revealed an increase in LALGLE (−0.173288587). The lesion kept growing in the following months, a finding consistent with PD. This case represents a good example of how radiomics could intercept changes in the radiological images that are almost invisible to the human eye and predict the evolution of each lesion. The slight increase of LALGLE could be referred to as the slight enlargement of the central hypodense area (yellow dotted circle). In this case, the CT window was stressed for demonstrative purposes.

**Figure 4 cancers-14-00350-f004:**
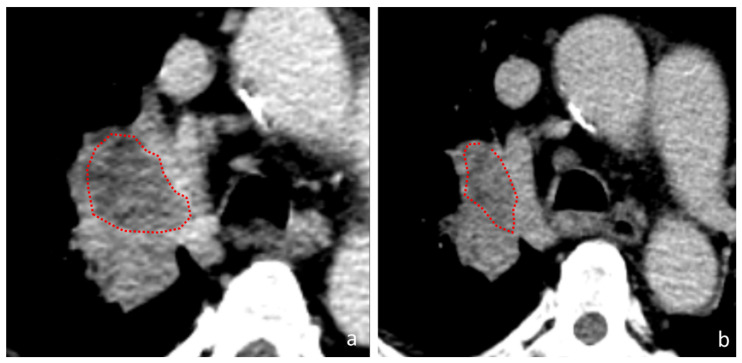
A tumor in the right pulmonary hilum. (**a**) baseline CT-scan (**b**) first assessment CT-scan. During follow-up, the tumoral mass shrunk a variation consistent with PR. At the first re-evaluation the central hypodense area reduced (red dotted circles); this tumor presented a ∆-LALGLE of 1.487953173, which consists of a reduction of 149% of the low attenuation area.

**Figure 5 cancers-14-00350-f005:**
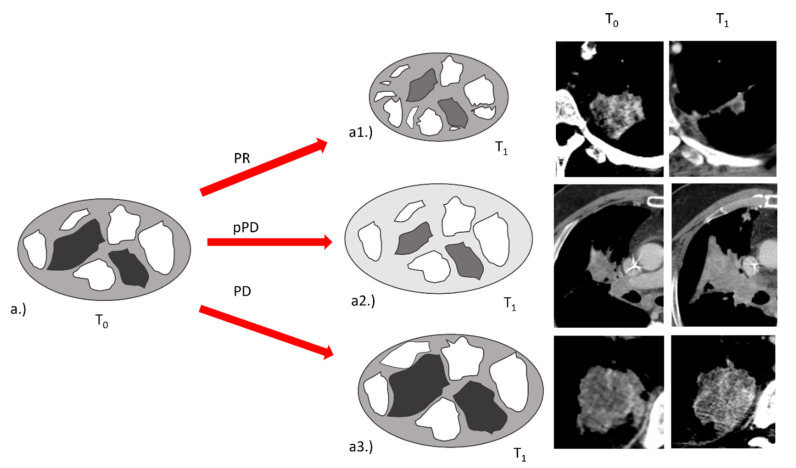
Exemplification of the variations of LAHGLE and LALGLE after the introduction of immunotherapy. (**a**) At baseline a tumor presented with different zones of enhancement (white) and low attenuation (dark grey) within the tumor (light grey). (**a1**) In PR, the tumor shrinks and the values of LAHGLE and LALGLE decrease, indicating a reduction in dimension of high and low attenuation “large zones”. (**a2**) In pPD, the tumor enlarges and the values of LAHGLE increase but LGLZE decrease, indicating a reduction of low attenuation zones on a global level (the background and the low attenuation zones become lighter) and enlargement of high attenuation zones. In pPD LALGLE is not different from PD, but the difference in LGLZE is consistent, meaning that large areas are not exclusively affected, but the process that involves the reduction of low attenuation pixels occurs on a global instead of a local level. (**a3**) In PD, the tumor enlarges and the values of LAHGLE and LALGLE increase, indicating a growth of high and low attenuation “large zones”.

**Figure 6 cancers-14-00350-f006:**
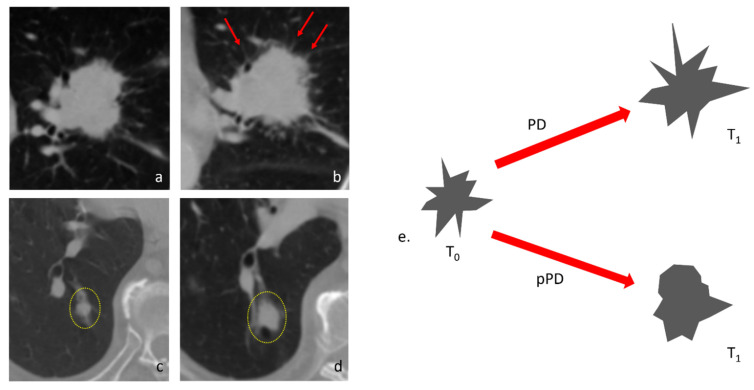
Examples of changes in Short Run Low Gray-Level Emphasis. (**a**) A tumor in the left upper lobe that undergone PD after immunotherapy (**b**). At the first re-evaluation, many spicules are appreciable on the boundaries of the tumor (red arrows), indicating a more infiltrative patterns, In this case, the value of SRLGLE presented a relative variation of −0.59617207. (**c**) A tumor in the right lower lobe (yellow dotted circle) presented a discrete enlargement after immunotherapy (**d**), a finding that might indicate PD; however, during follow-up the nodule shrunk, a behavior consistent with pPD. In this case, ∆-SRLGLE was 1.189415458 due to a decisive decrease of SRLGLE. (**e**) The changing of SRLGLE in PD and pPD is depicted in this scheme with exemplificative purposes: in PD the spiculae on the boundaries of tumor increase significantly and contribute to creating new small, low-density interfaces that increase the value of SRLGLE. In pPD, despite the volumetric enlargement of the nodule, the boundaries become more rounded, reducing the presence of small interfaces.

**Figure 7 cancers-14-00350-f007:**
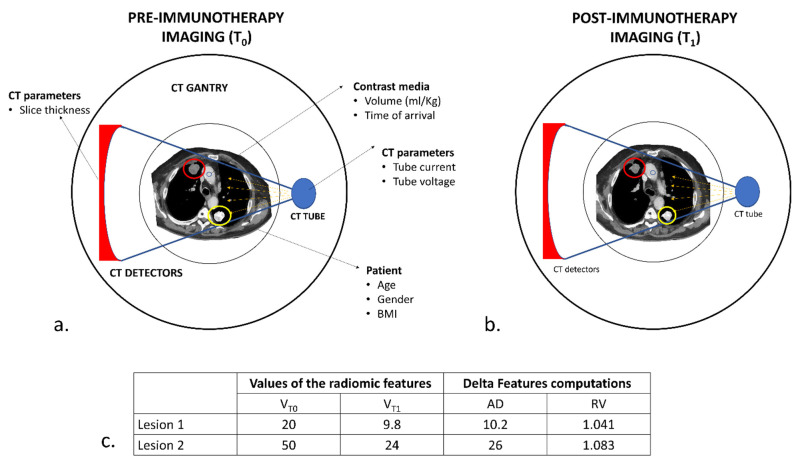
Hypothetical schematic representation of the variability in CT acquisition and delta-radiomic computations. (**a**) Baseline CT (T_0_) in a patient with two pulmonary lesions. Several factors might influence CT acquisition, depending on the type of scanner (number and dimensions of the detectors), the acquisition parameters employed (tube current and voltage) and the characteristics of the patient (age, gender and body mass index) from which depends, to some extent, the contrast medium kinetic and distribution. Even the position within the body of the pulmonary lesion might affect the radiomic analysis, as the values of the pixels depending on the attenuation of x-rays. Lesion 2 (yellow circle) lies between two high-attenuated bony structures (*), the spine and the scapula; lesion 1 (red circle), on the other hand, lies nearer to the skin and distant from the ribs or sternum. An x-ray beam will interact with these two lesions differently, resulting in different values of radiomic features. (**b**) At the first re-evaluation (T_1_), the two lesions responded to immunotherapy. In this case, each lesion will be acquired in conditions very similar to T_0_. (**c**) During delta-radiomic features computation, variations due to CT acquisition are eliminated using the RV method, which can extrapolate a similar change in different situations. We can appreciate how the two lesions, despite their differences in values and AD-calculated delta features, presented comparable changes.

**Table 1 cancers-14-00350-t001:** Delta-radiomic features were calculated with the RV method that resulted statistically different in the three response classes. The results are presented as median values for each class and *p*-values.

Relative Variation					
Radiomic Features		Response			
Class	Name	PR	SD	PD	*p*-Value
**firstorder**	Median	0.07389	0.04288	−0.2143	0.0072
	Energy	0.7002	−0.02225	−0.3811	0.0071
	TotalEnergy	0.8288	−0.1987	−0.4173	0.0038
	90Percentile	0.1130	0.0006320	−0.07377	0.02
**gldm**	DependenceNonUniformity	0.6793	−0.08826	−0.2910	0.0034
	GrayLevelNonUniformity	0.7474	−0.1150	−0.2443	0.0109
	SmallDependenceEmphasis	−0.2645	0.1324	0.1725	0.0165
**glcm**	Contrast	−0.2338	0.07732	0.3687	0.0166
	DifferenceEntropy	−0.08534	0.01756	0.04661	0.0242
	DifferenceVariance	−0.2241	0.04970	0.3384	0.006
	DifferenceAverage	−0.1293	0.04207	0.07722	0.0284
**glrlm**	RunVariance	0.1638	−0.05781	−0.1424	0.0419
	GrayLevelNonUniformity	0.8304	−0.09103	−0.2589	0.0107
	LongRunEmphasis	0.1187	−0.05526	−0.08364	0.0185
	RunLengthNonUniformity	0.5123	−0.1132	−0.3272	0.0036
	ShortRunEmphasis	−0.03467	0.01885	0.01652	0.0199
	RunPercentage	−0.04131	0.02180	0.02883	0.0329
	RunLengthNonUniformityNormalized	−0.06849	0.04196	0.03199	0.0414
**glszm**	ZoneVariance	1.531	−0.2942	−0.5456	0.01
	GrayLevelNonUniformity	0.1498	−0.02237	−0.2220	0.0147
	LargeAreaEmphasis	1.530	−0.3100	−0.5455	0.011
	ZonePercentage	−0.3220	0.1571	0.2031	0.0188
	LargeAreaLowGrayLevelEmphasis	2.474	−0.1821	−0.6068	0.016
	LargeAreaHighGrayLevelEmphasis	2.114	−0.1915	−0.2531	0.0295
**ngtdm**	Coarseness	−0.4023	0.06068	0.2851	0.0062
	Strength	−0.4111	0.3760	0.6139	0.0259
	Busyness	0.8051	−0.1563	−0.3590	0.0205

**Table 2 cancers-14-00350-t002:** Delta-radiomic features were calculated with the RV method that resulted statistically difference in PD and pPD lesions. The results are presented as median values for each class and *p*-values.

Relative Variation				
Radiomic Features		Response		
Class	Name	PD	pPD	*p*-Value
**gldm**	DependenceEntropy	0.001323	−0.05896	0.0072
	LowGrayLevelEmphasis	−0.03248	1.224	0.0036
**glcm**	Idmn	−0.0008908	−0.01153	0.05
	Idn	0.0002237	−0.02028	0.0357
**glrlm**	ShortRunLowGrayLevelEmphasis	−0.02426	1.454	0.0036
	LowGrayLevelRunEmphasis	−0.05360	1.352	0.0071
	RunLengthNonUniformity	−0.3272	−0.7698	0.0328
**glszm**	GrayLevelNonUniformity	−0.2220	−0.7201	0.0346
	LowGrayLevelZoneEmphasis	0.03821	2.818	0.0199

## Data Availability

Data is contained within the article or Appendix A.

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
