# Peer review of "Exploring Response to Immunotherapy in Non-Small Cell Lung Cancer Using Delta-Radiomics"

_cancers, 2022, doi:10.3390/cancers14020350_

Round 1
Reviewer 1 Report
Delta radiomics technique was employed to study the response of immunotherapy treatment in non-small cell lung cancer. The digital images obtained from CT scan were used for this analysis. The authors compared the CT scan data (pre-, post-immunotherapy) and used them as biomarkers. If successful, this can be a good technique for non-invasive cancer prognosis. Following are my comments on this manuscript.
- Most of the results are in a discussion part whereas only two tables (without legends and description) are in a result part. The authors mixed results together with discussion. It is very difficult to follow and understand the context. The authors should re-structure this manuscript. This is a serious flaw for the manuscript.
- The result, especially in table 1 and 2 , has not been described well. The authors should explain more in details in both tables.
- The authors should compare and confirm their results with some conventional predictive and prognostic markers for NSCLC. Generally, patients who went through immunotherapy treatment, should have predictive or prognostic test data.
- Human subjects were involved in this study, the institutional review board statement shall be declared.
- In methods sections, diagrams (Fig1 and 2) helps to understand the research plan but it is still difficult to follow. More description in methods could help.
Author Response
Delta radiomics technique was employed to study the response of immunotherapy treatment in non-small cell lung cancer. The digital images obtained from CT scan were used for this analysis. The authors compared the CT scan data (pre-, post-immunotherapy) and used them as biomarkers. If successful, this can be a good technique for non-invasive cancer prognosis. Following are my comments on this manuscript.
Most of the results are in a discussion part whereas only two tables (without legends and description) are in a result part. The authors mixed results together with discussion. It is very difficult to follow and understand the context. The authors should re-structure this manuscript. This is a serious flaw for the manuscript.
Thanks for the comments. We have made the following changes:
1) we have inserted the captions of the tables.
2) Yes, the text was hard to follow. We re-arranged the “Methods”, “Results” and “Discussion” sections to improve the clarity of the text. Moreover, the “Results” section was divided into specific paragraphs for each class of radiomic features.
The result, especially in table 1 and 2 , has not been described well. The authors should explain more in details in both tables.
Thank you for your suggestion. we apologize for these oversights. We have included an explanatory caption for both Table 1 and Table 2
Table 1: Delta-radiomic features calculated with the RV method that resulted statistically different in the 3 response classes. The results are presented as median values for each class and p-values
Table 2. Delta-rRadiomic ∆features calculated with the RV method that resulted statistically different in PD and pPD lesions. The results are presented as median values for each classes and p-values.
The authors should compare and confirm their results with some conventional predictive and prognostic markers for NSCLC. Generally, patients who went through immunotherapy treatment, should have predictive or prognostic test data.
Thank you for your suggestion. We performed univariate analysis considering the main prognostic and predictive factors in NSCLC: age, ECOG PS, histology and PD-L1 expression. In the univariate analyzes, none demonstrated to be able to discriminate the patient's prognosis in a statistically significant way. Unfortunately, the PD-L1 expression data were present only in patients treated with pembrolizumab in the first line, while they were not available for the majo to rity of patients treated with nivolumab in the second line (in the two patients of which it was available, it was higher than 50 %). For these reasons it was not possible to perform a univariate analysis which would have been affected by a selection bias (the patients whose expression we do not have, are all treated in the second line and therefore with a predictably worse prognosis). We have included an explanatory subsection in the paragraph of methods, results and discussion.
Human subjects were involved in this study, the institutional review board statement shall be declared.
we apologize for this oversight. We have inserted the Institutional Review Board Statement: The study was conducted according to the guidelines of the Declaration of Helsinki, and approved by the Ethics Committee of Ospedale Policlinico San Martino (protocol code CER Liguria 191/2015 – id: 2542 and date of approval of the last amendment 09 september 2021.”
In methods sections, diagrams (Fig1 and 2) helps to understand the research plan but it is still difficult to follow. More description in methods could help
Thank you for your suggestion we added the following description in the text.
For Figure 1 from line 90 to 93: Sixty-two patients were enrolled in the study, from these, 24 patients were excluded due to lack of reevaluation CT (early death, complete response of target lesion, absence of contrast medium in reevaluation CT). In addition, for 5 patients there were problems with reformatting of the slices. Therefore only 33 patients were considered and in particular 43 overall target lesions.
Furthermore we have modified the description of the figure 2 as follow: Although the radiomic analysis aimed at the study included the exclusive analysis of the baseline and the first re-evaluation CT scans, in order to generate a predictive model of response, we evaluated the response in the subsequent CT scans performed throughout the patient's therapy. Finally, we assigned the best response available in subsequent patient evaluation (a SD at the first re-evaluation that became a PR after other re-evaluations was considered a PR). For the same reason, in case of disease progression, a SD at the first re-evaluation was considered a PD (Figure 2).
Reviewer 2 Report
General comments:
The structure of the paper is quite disorganised, as several paragraphs belong to other sections rather than to the sections they are currently located. A large amount of information that is in Discussion should be in Methods and Results. Also, important methodological details appear for the first time in Discussion.
Methods/results: Details about the machine learning technique used in this study are lacking. If no machine learning algorithms were employed, this should be mentioned in the Methods.
It is not clear how the radiomic features were identified/selected that were further used for delta radiomics.
Section 3.1 would better fit in the Methods section than in Results.
Discussion: The number of patients is very small and mixed for a radiomic analysis. The discussion section must include a more detailed paragraph on the limitations of this work and explain the impact of the low number of analysed CT scans/patients on the final results.
Specific comments:
Line 78 – to individuate radiomic features to be used….
Line 94 – remove ‘obviously’
Figure 1 – 24 patients did not have the first…. ( ….early death…)
Figure 2 – the caption must be changed/adapted to the chart. It should not start with ‘we used …’
Line 129 – formula is missing after section 2.4 for delta radiomics
Line 137 – remove initials of the radiologist
Figure 3 – this is not a figure, they are equations. T1 should be subscript instead of multiplication. This part should appear at Methods, where the delta radiomic analysis is explained and not in Discussion.
Line 203 – rephrase the sentence starting with ‘Hypothetically…’
Author Response
The structure of the paper is quite disorganised, as several paragraphs belong to other sections rather than to the sections they are currently located. A large amount of information that is in Discussion should be in Methods and Results. Also, important methodological details appear for the first time in Discussion.
Thanks for your suggestions. We agree with the reviewers that the text was hard to follow; we rearranged the text to improve clarity.
Methods/results: Details about the machine learning technique used in this study are lacking. If no machine learning algorithms were employed, this should be mentioned in the Methods.
In this study, we did not employ a machine learning technique, so we have added this sentence in the “statistical analysis” subsection in the methods paragraph : “Due to the limited dataset, no machine learning or other type of artificial intelligence approach was adopted.”
Usually, radiomic features are processed with complex mathematical models and/or machine learning that permit to perform classification tasks. However, in this work we chose classical statistical methods over machine learning for several reasons:
- The small size of the classes could generate serious bias in the study.
- Delta features are not comparable to standard radiomic features, which typically are highly correlated with each other, a fact that preclude the employment of technique such as multivariable analysis. Delta features are derivative of standard radiomic features and their value is that they can describe the changing of a definite situation in time.
- Adopting a machine learning approach in this study would have made very difficult the semantical interpretation of the delta features. Machine learning necessitates that the features are pre-processed using technique such as Principal Component Analysis (PCA). In PCA, radiomic features are variably assembled into complex descriptors called principal component that summarize the most critical aspect of several features; in this process, the essential pieces of information are preserved, and redundant ones are eliminated. However, principal components can be hardly interpreted to elaborate interpretations such those described in the paper.
A specific paragraph about machine learning was already present in the discussion, highlighting the fact that a larger cohort is needed to properly process data using such a technique. We think that machine learning will be essential in future studies but in this one we chose to focus on the semantical meaning of the features.
It is not clear how the radiomic features were identified/selected that were further used for delta radiomics.
Yes, this part was not clear, thank you for your suggestion. The Pyradiomics software permits the extraction of 93 features using the following methods: first-order statistics, Gray Level Co-Occurrence Matrix, Gray Level Run Length Matrix, Gray Level Size Zone Matrix, Neighbouring Gray Tone Difference Matrix and Gray Level Dependence Matrix. All the features were processed through delta radiomics. Specific paragraphs regarding these aspects were added to the text.
Section 3.1 would better fit in the Methods section than in Results.
Thanks you very much for your suggestion. We have been discussing the possibility of moving this paragraph as you suggested. In consideration of the current structure of the paper and the contents of the paragraph, we considered it more correct to maintain the current position. In fact, in the paragraph of methods there is already a paragraph on the inclusion criteria for patients and how they were selected, while the paragraph dedicated to patients in the results section reports the characteristics of the patients included and therefore formally fit within the results of the work. Despite this, we are grateful for the comment that allowed us to rethink the structure of the paper.
Discussion: The number of patients is very small and mixed for a radiomic analysis. The discussion section must include a more detailed paragraph on the limitations of this work and explain the impact of the low number of analysed CT scans/patients on the final results.
Thank you for your comment. We have modified the text in order to better explain the limitations of the study.
Specific comments:
Line 78 – to individuate radiomic features to be used….
Line 94 – remove ‘obviously’
Figure 1 – 24 patients did not have the first…. ( ….early death…)
Figure 2 – the caption must be changed/adapted to the chart. It should not start with ‘we used …’
Line 129 – formula is missing after section 2.4 for delta radiomics
Line 137 – remove initials of the radiologist
Figure 3 – this is not a figure, they are equations. T1 should be subscript instead of multiplication. This part should appear at Methods, where the delta radiomic analysis is explained and not in Discussion.
Line 203 – rephrase the sentence starting with ‘Hypothetically…’
Thank you for your comments, we have corrected all the typos that you suggest.
Round 2
Reviewer 1 Report
The authors addressed all the questions.
Author Response
Thank you very much
Reviewer 2 Report
The authors have improved the scientific content / flow of the manuscript by considering most of the comments. There are, however, some minor issues that were not addressed:
- Figure 1 – replace ‘24 patients have not the first…’ with ‘24 patients did not have the first…’
- Discussion – line 330 – remove ‘Figure 3’
- The two equations must be rewritten in the equation editor; they should not look like an image
The language should be revised throughout the manuscript, as there are still several linguistic errors that require corrections.
Author Response
The authors have improved the scientific content / flow of the manuscript by considering most of the comments. There are, however, some minor issues that were not addressed:
- Figure 1 – replace ‘24 patients have not the first…’ with ‘24 patients did not have the first…’
Thank you very much for your comment.We have corrected the sentence in the figure.
- Discussion – line 330 – remove ‘Figure 3’
Thank you for your comment. We have removed 2(figure 3)”
The two equations must be rewritten in the equation editor; they should not look like an image
Thank you for your comment. We have rewritten the two equation with the equation editor.
The language should be revised throughout the manuscript, as there are still several linguistic errors that require corrections.
thanks for your comment, we have revised the english language